# The Transcription Factors *WRKY41* and *WRKY53* Mediate Early Flowering Induced by the Novel Plant Growth Regulator Guvermectin in *Arabidopsis thaliana*

**DOI:** 10.3390/ijms24098424

**Published:** 2023-05-08

**Authors:** Chenyu Yang, Chongxi Liu, Shanshan Li, Yanyan Zhang, Yi Zhang, Xiangjing Wang, Wensheng Xiang

**Affiliations:** 1The State Key Laboratory for Biology of Plant Disease and Insect Pests, Institute of Plant Protection, Chinese Academy of Agricultural Sciences, Beijing 100193, China; yangchenyu0825@163.com (C.Y.); ssli@ippcaas.cn (S.L.); yyzhang@ippcaas.cn (Y.Z.); 2Key Laboratory of Agriculture Biological Functional Gene of Heilongjiang Provincial Education Committee, Northeast Agricultural University, No. 600 Changjiang Street, Xiangfang District, Harbin 150030, China; liuchongxi@neau.edu.cn; 3Department of Biology, School of Life Sciences, Institute of Plant and Food Science, Southern University of Science and Technology, Shenzhen 518055, China; zhangy66@mail.sustech.edu.cn

**Keywords:** guvermectin, flowering, *WRKY41*, *WRKY53*, *SOC1*, *LFY*

## Abstract

Flowering is a crucial stage for plant reproductive success; therefore, the regulation of plant flowering has been widely researched. Although multiple well-defined endogenous and exogenous flowering regulators have been reported, new ones are constantly being discovered. Here, we confirm that a novel plant growth regulator guvermectin (GV) induces early flowering in *Arabidopsis*. Interestingly, our genetic experiments newly demonstrated that *WRKY41* and its homolog *WRKY53* were involved in GV-accelerated flowering as positive flowering regulators. Overexpression of *WRKY41* or *WRKY53* resulted in an early flowering phenotype compared to the wild type (WT). In contrast, the *w41/w53* double mutants showed a delay in GV-accelerated flowering. Gene expression analysis showed that flowering regulatory genes *SOC1* and *LFY* were upregulated in GV-treated WT, *35S:WRKY41*, and *35S:WRKY53* plants, but both declined in *w41/w53* mutants with or without GV treatment. Meanwhile, biochemical assays confirmed that *SOC1* and *LFY* were both direct targets of WRKY41 and WRKY53. Furthermore, the early flowering phenotype of *35S:WRKY41* lines was abolished in the *soc1* or *lfy* background. Together, our results suggest that GV plays a function in promoting flowering, which was co-mediated by WRKY41 and WRKY53 acting as new flowering regulators by directly activating the transcription of *SOC1* and *LFY* in *Arabidopsis*.

## 1. Introduction

Flowering is a key agronomic trait that plays an essential role during plant growth and development; it is also a major signal for the developmental transition from vegetative to reproductive growth [1]. Studies have shown that flowering is affected by multiple environmental conditions and endogenous developmental cues [1,2,3,4,5,6]. Over recent years, the molecular mechanisms and genetics of flowering have been reviewed in detail, and the main flowering pathways, such as vernalization, photoperiod, autonomous, age, and gibberellin (GA) pathways, have been revealed to explain flowering signaling in the model plant *Arabidopsis thaliana* [2,7]. Although flowering is affected by a variety of factors, genetic studies have shown that these flowering pathways converged on some key flowering regulatory genes, including *CONSTANS* (*CO*), *FLOWERING LOCUS T* (*FT*), *TWIN SISTER OF FT* (*TSF*), *FLOWERING LOCUS C* (*FLC*), *SUPPRESSOR OF OVEREXPRESSION OF CO1* (*SOC1*), *APETALA1* (*AP1*), and *LEAFY* (*LFY*), to regulate flowering in plants [8,9,10,11,12].

In *Arabidopsis*, *SOC1* acts as a key floral integrator gene. It encodes a MADS-box transcription factor that responds to multiple flowering pathways [13,14]. In the vegetative phase, *SOC1* expression is suppressed by *FLC* and *SHORT VEGETATIVE PHASE* (*SVP*), but it can be induced by *FT* and GA in the floral transition stage [15,16,17]. *SOC1*, in collaboration with *AGL24*, can also regulate the floral meristem identity gene *LFY* by directly binding to its promoter [18]. *SOC1* was also shown that be upregulated by the WRKY genes in an unknown way, such as *AtWRKY75* [4]. *LFY* is an important flowering-time gene and plays a crucial role in determining flowering time [19,20]. *LFY* expression is rapidly increased upon floral induction, and it can be regulated by other genes, such as *AtWRKY71* [21]. Meanwhile, *LFY* has also been proven to directly regulate the expression of the downstream gene *AP1* [22].

Previous studies have shown that WRKY transcription factors (TFs) are involved in various physiological processes and play important roles in plant growth and development [23,24]. WRKY TFs are a type of DNA-binding protein identified by the peptide WRKYGQK at the N-terminus and a zinc finger motif at the C-terminus [25]. More than 70 WRKY genes have been found in *Arabidopsis thaliana*, and extensive molecular biology studies have shown that they are involved in plant growth and development, and biotic and abiotic stress signal transduction [23,26,27,28,29]. For instance, *AtWRKY8*, *AtWRKY33*, *AtWRKY38*, *AtWRKY62,* and *AtWRKY76* are involved in the response to fungal pathogens and abiotic stress [30,31,32,33]. AtWRKY71 interacts with EXB1 to control shoot branching by regulating RAX genes [34]. Moreover, it has been confirmed that WRKY genes, such as *AtWRKY6*, *AtWRKY71*, *GsWRKY20*, and *OsWRKY11*, positively regulate plant flowering [21,35].

Plant growth regulators (PGRs) play important roles in plant growth, development, and stress resistance, and have been widely used in agricultural production [36,37,38,39,40,41]. PGRs can also directly or indirectly affect plant flowering by acting as repressors or activators [9,10,42,43]. For example, brassinosteroid (BR) promotes the expression of the floral repressor *FLC* and its homologs to result in delayed flowering [44]. Physiological evidence indicates that cytokinin plays a role in floral transition and promotes flowering by activating the transcription of *TSF* and *SOC1* [5]. Gibberellin (GA) plays multiple functions in plant development and promotes flowering by upregulating the floral meristem identity gene *LFY*, which is necessary for flower formation [9]. Yu et al. (2012) demonstrated that GA regulates floral transition through DELLA interacting directly with *SQUAMOSA PROMOTER BINDING-LIKE* (*SPL*).

Guvermectin (GV) is a novel N9-glucoside cytokinin compound identified from *Streptomyces caniferus NEAU6* and has been successfully registered as a novel natural PGR (Registration Code: PD20212929) in China [45,46]. Notably, although GV is a nucleoside analog like cytokinin, the cytokinin receptor triple mutant *ahk2-2ahk3-3cre1-12* still responds to GV treatment, indicating that GV plays a novel mechanism different from that of cytokinin [47]. Recently, the significant biological activity of GV in regulating plant growth and development has been confirmed. GV was shown that plays a role in promoting maize growth in high-temperature environments [45]. Moreover, GV also promoted root and hypocotyl growth in Arabidopsis and seed germination, tillering, and early maturing in rice [47]. Interestingly, we found that GV also has an effect in inducing early flowering in plants. However, little was known about the mechanism of GV acts in regulating plant flowering. Our biochemical and genetic experiments further demonstrated that the WRKY genes *WRKY41* and *WRKY53* are induced by GV and they co-mediate GV-accelerated flowering by directly binding to the *SOC1* and *LFY* promoters and activating transcription in *Arabidopsis*. Meanwhile, this study provides a molecular basis for the application of GV to plants.

## 2. Results

### 2.1. The Plant Regulator GV Can Accelerate Flowering 

In our previous study, the significant biological activity in regulating plant growth and development of PGR guvermectin (GV) has been identified [45,47]. Moreover, in our biological function assays, GV was shown to have an effect in promoting flowering in *Arabidopsis*. GV (50 mg L^−1^) [45] was sprayed on two-week-old, wild-type (WT) plants grown under long-day (LD) conditions (16 h light/8 h dark) and treated with 0 mg L^−1^ GV as a control. The same treatment was performed on seven-week-old *Arabidopsis* grown under short days (SD) conditions (8 h light/16 h dark). As shown in Figure 1, GV treatment, significantly accelerated flowering (Figure 1A and Appendix A) compared to the control which was treated with 0 mg L^−1^ GV, as measured by days to flowering (DTF) (Figure 1B) and rosette leaves number (RLN) (Figure 1C). Similarly, GV also significantly accelerated *Arabidopsis* flowering under SD conditions (Appendix A and Appendix A). The cytokinin receptor triple mutant *ahk2-2 ahk3-3 cre1-12* showed a response to GV [47], moreover, three cytokinin receptor mutants *ahk2/3*, *ahk2/4*, and *ahk3/4* all showed a response to GV and flowering early after GV treatment compared to the control (Appendix A), suggesting GV is different from cytokinin in regulating flowering. Together, these results suggest that GV accelerates flowering with a new mechanism in *Arabidopsis*.

### 2.2. WRKY41 Is Significantly Upregulated by GV

To investigate the mechanism by which GV accelerates flowering in plants, RNA sequencing (RNA-seq) was performed on *Arabidopsis* plants four days after GV treatment. The plants were treated with 0 mg/L GV as a control. Our analysis’s results showed that 1358 genes were significantly induced to express by GV (Appendix A and Dataset S1), meanwhile, the results of multiple types of gene qPCR consistent with RNA-seq verified the reliability of the data (Appendix A). TFs were selected for analysis because of their important roles in signaling pathways controlling plant growth [24,48]. The transcriptome data analysis showed that multiple classes of TFs were affected by GV treatment, interestingly, WRKY TFs showed the largest change in transcript abundance (log_2_FoldChange *≥* 1.5) (Appendix A and Appendix A). Notably, WRKY TFs have aroused our attention because they were previously reported to be involved in plant flowering, such as *AtWRKY71* [21]. To further confirm these results, the expression levels of selected WRKY genes (Appendix A) were determined by quantitative PCR (qPCR) at 0, 1, 3, 5, and 7 d after GV treatment, respectively. The results showed that the selected WRKY genes were induced at different levels by GV. WRKY41 was the most highly upregulated in response to GV (Figure 2A) of the WRKY genes studied here (Appendix A–K). WRKY41 is reportedly involved in plant growth and development and is expressed in floral organs [49], implying that WRKY41 may play a role in flowering development. Thus, the significant induction of WRKY41 by GV suggested the possibility that WRKY41 involves in the process of GV-accelerated flowering.

### 2.3. WRKY41 and Its Homolog WRKY53 Play Roles in GV-Accelerated Flowering

To determine the biological functions of *WRKY41* in regulating flowering, we generated two overexpression of *WRKY41* (*35S:WRKY41*) lines (Figure 2(Bi) and Appendix A) and obtained the knockout mutant *wrky41* (Figure 2(Bii) and Appendix AD,E). The results showed that *35S:WRKY41* lines flowered early compared to WT plants (Figure 2C–E and Appendix A), suggesting that *WRKY41* has a role in flowering regulation. Notably, *WRKY61* showed a significantly elevated upon GV induction (Appendix A), However, the overexpression of *WRKY61* does not affect flowering (Appendix A) indicating that it might not be involved in the regulation of flowering. In addition, the flowering phenotypes were indistinguishable between *wrky41* and WT plants (Figure 2F–H), implying that other WRKY genes must be involved. Importantly, GV-induced early flowering was attenuated in the *wrky41* mutants compared with that in WT plants (Figure 2G,H), suggesting that *WRKY41* contributes to the process of GV-accelerated flowering.

A phylogenetic tree of WRKY TFs indicated that *WRKY53* is the closest homolog to *WRKY41* (Appendix A), consistent with the description of Wu et al. [50]. The RNA-seq and qPCR data both showed that *WRKY53* levels were altered after GV treatment (Appendix A and Figure 3A), suggesting that *WRKY53* was induced by GV. First, to analyze the function of WRKY53 in the regulation of flowering, two overexpression of *WRKY53* (*35S:WRKY53*) lines (Figure 3(Bi) and Appendix A) were generated and the knockout mutant *wrky53* (Figure 3(Bii) and Appendix A) was obtained to analyze the flowering phenotypes. The *35S:WRKY53* lines exhibited earlier flowering than WT plants (Figure 3C–E and Appendix A), suggesting that *WRKY53* plays a role in regulating flowering. However, no differences in flowering phenotypes of *wrky53* and the WT plants (Figure 3F–H and Appendix A) implied that still other WRKY genes must be involved. Interestingly, similar to the results of *WRKY41*, the GV-induced early flowering phenotype was also weakened in *wrky53* compared to that in WT plants (Figure 3G,H). This suggested that *WRKY41* and *WRKY53* may be functionally redundant. We therefore further explored the potential redundant function of *WRKY41* and *WRKY53* in regulating flowering. A double knockout mutant was generated for *WRKY41* and *WRKY53* using CRISPR/Cas9-mediated genome editing technology. Two homozygous lines, *w41/w53-1* and *w41/w53-2*, were confirmed by sequencing (Figure 4A,B). Both double mutant lines flowered later than the WT plants (Figure 4C–E and Appendix A), indicating that *WRKY41* and *WRKY53* indeed have redundant functions in regulating flowering. Notably, the GV-induced early flowering phenotypes in WT were not present in the *w41/w53* mutants (Figure 4C–E), implying that *WRKY41* and *WRKY53* play important roles in the process of GV-accelerated flowering. Taken together, these results suggest that *WRKY53* functions redundantly with *WRKY41* and they co-mediate GV-accelerated flowering.

### 2.4. WRKY41 and WRKY53 Activate the Transcription of SOC1 and LFY

The biological activity of GV in accelerating flowering has previously been confirmed. However, it remains to be verified whether flowering regulatory genes are involved in the process of GV flowering induction. Our transcriptome data analysis showed that many flower-regulated genes are differentially expressed after GV treatment (Appendix A). During flowering, the major flowering regulatory genes *GI*, *CO*, *FT*, *SOC1*, *AP1, LFY*, *FLC*, and *TFL1* are regulated [9,14,51]. We collected samples at 0, 1, 3, 5, and 7 d after 50 mg L^−1^ GV and control (0 mg L^−1^ GV) treatment. Our results showed the transcript levels of the floral regulatory genes *SOC1*, *LFY,* and *AP1* were significantly upregulated (2~6 foldchange) compared to the control after GV treatment in *Arabidopsis* (Figure 5A–C). *GI*, *CO*, *FT*, *FLC*, and *TFL1* showed only weak upregulation at some points (Appendix A–F), suggesting that *SOC1*, *AP1,* and *LFY* are the key flowering regulators induced by GV. In the *35S:WRKY41* and *35S:WRKY53* lines, the related transcript levels of *SOC1*, *LFY,* and *AP1* were significantly elevated compared to WT plants (Figure 5D–F), indicating that they are the primary flowering regulators induced by WRKY41 and WRKY53. To further verify whether WRKY41 and WRKY53 have effects on the transcriptional activation of *SOC1*, *LFY*, and *AP1*, a dual-luciferase (Luc)-based reporter assay was conducted in *Nicotiana benthamiana*. The WRKY41 and WRKY53 proteins acted as the effectors and the 2-kb promoter regions of *SOC1*, *LFY*, and *AP1* were the reporters. We found that in the presence of WRKY41 or WRKY53, the expression of luciferase driven by the native *SOC1* and *LFY* promoters was greater than that driven by the same promoters carrying mutated W-boxes (Figure 5G–J and Appendix A), but *AP1* was not activated (Appendix A), suggesting that WRKY41 and WRKY53 mainly regulate the expression of *SOC1* and *LFY*. *SOC1* and *LFY* expression levels were also measured in the *w41/w53* lines via qPCR. The results showed that *SOC1* and *LFY* are down-regulated in the *w41/w53* lines compared to the WT with and without GV treatment, respectively (Figure 5K,L), again showing regulation of *SOC1* and *LFY* by WRKY41 and WRKY53, respectively. In addition, *SOC1* and *LFY* were detected at low levels in the *w41/w53* lines compared to that in WT after GV treatment, indicating that the upregulation of *SOC1* and *LFY* by GV is mediated by *WRKY41* and *WRKY53*. Together, these results indicated that WRKY41 and WRKY53 mediate GV-induced expression of *SOC1* and *LFY*.

### 2.5. WRKY41 and WRKY53 Directly Bind to the Promoters of SOC1 and LFY 

The findings that WRKY41 and WRKY53 mediate GV-induced *SOC1* and *LFY* expression prompted us to investigate the relationship between the two transcription factors and the two flowering regulatory genes. We first tested whether WRKY41 and WRKY53 directly regulate *SOC1* and *LFY* transcription by binding to the promoter regions. Analysis of the 2-kb promoter regions of *SOC1* and *LFY* showed that they contained one and six W-box elements, respectively (Figure 6A). We then performed an electrophoresis mobility shift assay (EMSA) to confirm in vitro interactions between GST-tagged WRKY41 and WRKY53 proteins and 200-bp probes containing the W-box elements found in the promoters of *SOC1* and *LFY*. A 200-bp probe without the W-box elements was used as a control. It was evident that both the GST-WRKY41 and GST-WRKY53 proteins could strongly bind to the W-box element in the *SOC1* promoter (Figure 6B and Appendix A), and they were also shown to strongly bind to the W-box elements in the *LFY* promoter (Figure 6C and Appendix A). This indicated that WRKY41 and WRKY53 were able to directly bind to the promoters of *SOC1* and *LFY* through the W-box elements.

We next performed chromatin immunoprecipitation (ChIP)-qPCR analysis using the transgenic *35S:WRKY41*-GFP and *35S:WRKY53*-GFP lines to determine whether WRKY41 and WRKY53 could directly bind to the *SOC1* and *LFY* promoters in vivo. Consistent with the results in vitro, WRKY41 and WRKY53 were both bound to the W-box element in the *SOC1* promoter (Figure 6D and Appendix A); WRKY41 was bound to the LFY-1 and LFY-3 W-box element, and WRKY53 was bound to the LFY-1 W-box element in the *LFY* promoter (Figure 6E and Appendix A). These results suggest that WRKY41 and WRKY53 can directly bind to *SOC1* and *LFY* promoters, in vivo.

### 2.6. Mutations in SOC1 and LFY Suppress Early Flowering in 35S:WRKY41 Lines

The results described above suggested that WRKY41 and WRKY53 likely promote flowering by directly activating the transcription of *SOC1* and *LFY*. To determine the genetic relationship between WRKY41/WRKY53 and *SOC1*/*LFY* in flowering regulation, we generated *SOC1* and *LFY* knockout lines in both WT and *35S:WRKY41* (Appendix A) backgrounds using the CRISPR/Cas9-mediated genome editing system. Sequencing confirmed that homozygous *soc1*, *35S:WRKY41-soc1*, *lfy*, and *35S:WRKY41-lfy* lines were obtained (Figure 7A,B). The flowering phenotypes of *soc1* (Figure 7C,E,F) and *lfy* (Figure 7D–F) were similar to those of the previously published *soc1-2* [13] and *lfy-1* [52] lines, respectively, and later compared to WT, indicating that the selected editing sites in *SOC1* and *LFY* were effective. Consistent with our findings that *SOC1* and *LFY* were directly downstream of WRKY41, the early flowering phenotype caused by overexpression of *WRKY41* was fully repressed in the *35S:WRKY41-soc1* (Figure 7C,E,F and Appendix A) and *35S:WRKY41-lfy* lines (Figure 7D–F and Appendix A). These results suggested that the early flowering in 35*S:WRKY41* is mainly attributable to the induction of *SOC1* and *LFY* expression. Moreover, the flowering time of *soc1* and *lfy* lines could not be improved after GV treatment (Appendix A). These results revealed that WRKY41 mediates GV-induced flowering in a *SOC1*/*LFY*-dependent manner.

## 3. Discussion

Plant growth and development are affected by a variety of biotic and abiotic stresses. To ensure reproductive success and complete seed development under favorable natural conditions, controlled regulation of flowering has been considered an essential measure that is used in agriculture [7]. The promotion of plant flowering has been shown to protect plants against harsh environmental conditions, such as pathogen attacks, drought, heat, and frost, which endanger seed production and harvesting [53,54]. Recent studies have confirmed that plant flowering is affected by PGRs, for example, exogenous application of cytokinin (6-BA) promotes flowering in *Arabidopsis* [5]. Gibberellic (GA) played an important role in accelerating flowering [11,13,55]. In the present study, we confirmed the function of a novel PGR, guvermectin (GV), in inducing early flowering (Figure 1 and Appendix A). Although GV is a nucleoside analog like cytokinin, three cytokinin receptor mutants *ahk2/3*, *ahk2/4*, and *ahk3/4* all showed a response to GV and flowering early after GV treatment (Appendix A), suggesting GV is different from cytokinin and acts independent cytokinin signaling or downstream of cytokinin receptors in regulating flowering. Our findings suggested that GV could act as an important new exogenous factor to regulate plant flowering. Biochemical and genetic studies showed that *WRKY41* and *WRKY53* act as positive regulators of flowering and were induced by GV, and they were shown to co-mediate GV-accelerated flowering by directly activating the transcription of the flowering regulation genes *SOC1* and *LFY* in *Arabidopsis*.

Over the past decade, substantial progress has been achieved in defining the roles of WRKY TFs in various stress responses and plant development [19,23,24]. Accumulating data have shown that many WRKY genes are induced by and involved in PGRs-regulated plant growth and development. For example, *AtWRKY46*, *AtWRKY54,* and *AtWRKY70* are positively involved in BR-regulated growth in plants [46]. Auxin antagonizes leaf senescence through *AtWRKY57* [56]. Moreover, extensive studies have shown that WRKY genes, acting as positive or negative regulators, regulate flowering [4,19]. *AtWRKY75*, a positive regulator, accelerated flowering in *Arabidopsis* [4]. *AtWRKY12* and *AtWRKY13* modulate flowering time in opposite directions by directly targeting *FRUITFUL* (*FUL*) [19]. Here, an analysis of transcriptome data after GV treatment revealed that multiple WRKY genes were induced (Appendix A), and qPCR confirmed that those WRKY genes were induced at different levels in response to GV treatment (Appendix A–K). As the highly upregulated WRKY gene in response to GV treatment, *WRKY41* (Figure 2A) was selected for further study. A member of WRKY group III, *WRKY41* has been reported to play an important role in the regulation of plant growth and development, such as regulating seed dormancy [49]. Studies have shown *WRKY41* is expressed in different tissues of the plant, importantly, *WRKY41* is expressed in floral tissue [49], suggesting that it may play a role in flower development, but this needs to be confirmed. Here, we found that *WRKY41* did indeed function as a new flowering regulator. *35S:WRKY41* plants showed an early flowering phenotype compared to the WT (Figure 2C–E), providing evidence that *WRKY41* positively regulated flowering. However, the flowering time of *wrky41* was consistent with that of WT (Figure 2F–H), implying the existence of a functionally redundant WRKY gene. WRKY TFs have been shown to function redundantly in regulating flowering; for example, *AtWRKY71* is functionally redundant with two closely related homologs, *AtWRKY8* and *AtWRKY28*, in regulating flowering [21]. A constructed phylogenetic tree indicated that *WRKY53* was the closest homolog to *WRKY41* (Appendix A), interestingly, *WRKY53* was also upregulated after GV treatment in *Arabidopsis* (Figure 3A). *WRKY53* belongs to WRKY group III and has roles in regulating leaf senescence [57], plant disease resistance [58], and plant architecture and seed size in rice [59]. Moreover, *WRKY53* is involved in flowering through an unclear mechanism [21]. Our results showed that *35S:WRKY53* lines exhibited an early flowering phenotype compared to WT lines (Figure 3C–E), indicating that *WRKY53* also plays a role in flowering regulation. Notably, *w41/w53* double knockouts showed delayed flowering compared to WT (Figure 4C–E), revealing that *WRKY41* and *WRKY53* have redundant functions in flowering regulation. Previous studies have shown that WRKY TFs mediate PGRs-induced flowering. For example, the absence of *AtWRKY75* leads to a delay in GA-mediated flowering time [4]; *wrky12* mutants show less sensitivity to the GA-induced flowering response, but *wrky13* mutants are more sensitive to that [19]. Consistently, we found that *wrky41* and *wrky53* mutants had delays in GV-accelerated flowering time compared to WT plants (Figure 2F–H and Appendix A), indicating that *WRKY41* and *WRKY53* both play roles in the process of GV-accelerated flowering. Notably, the undifferentiated late-flowering phenotype of *w41/w53* lines with and without GV treatment (Figure 4C–E) provided evidence that *WRKY41* and *WRKY53* jointly mediate GV-accelerated flowering. We cannot exclude the possibility that other GV-responsive WRKY genes affect flowering; however, the loss of both WRKY41 and WRKY53 function suppressed GV-accelerated flowering (Figure 5), indicating that other WRKY genes may not be involved in this process. For example, we found the overexpression *WRKY61*, a WRKY gene responds highly to GV and showed no effect on flowering time (Appendix A), but the reduction of TCV viral accumulation in overexpression *WRKY61* lines indicates that WRKY61 mainly responds to stress [60]. This implies that GV would be able to induce a plant defense response–a promising topic for future study.

It has been reported that major flowering regulatory genes can be induced by a variety of factors that are dependent or not dependent on the flowering pathway in *Arabidopsis* [20,61,62]. It has also been confirmed that WRKY proteins regulate the expression of different floral integrators or floral meristem identity genes. For example, WRKY71 directly activates *FT* and *LFY* [21]; WRKY12 and WRKY13 directly regulate *FUL* [19]. WRKY75 directly activates *FT* [4]. We found that *SOC1*, *LFY,* and *AP1* were all significantly upregulated in GV-treated WT, *35S:WRKY41*, and *35S:WRKY53* plants (Figure 5A–F). Furthermore, a dual-luciferase reporter assay showed direct regulation of *SOC1* and *LFY* expression by WRKY41 and WRKY53 (Figure 5G–J and Appendix A). *SOC1* has a central role in the transition to flowering [14,62]. Studies have shown that *SOC1* is involved in regulating flowering time, floral patterning, and floral meristem (FM) determinacy. Furthermore, *SOC1* is not only induced by exogenous factors such as GA [63] and cytokinin [5] but can also be directly regulated by endogenous genes such as *FT* [64], *miR172* [65], and *NUCLEAR FACTOR Y* [13] through direct binding to its promoter. Although it has been demonstrated that *SOC1* can be regulated by a variety of factors, there was no prior evidence of a WRKY gene directly regulating *SOC1*. Interestingly, our EMSA and Chip-qPCR experiments demonstrated, for the first time, that WRKY41 and WRKY53 can directly bind to the *SOC1* promoter (Figure 6B,D and Appendix A). These results add WRKY genes to the list of known genes that directly regulate *SOC1* expression. *LFY* is a target gene of *SOC1* and is reportedly involved in regulating flowering time and FM determinacy [9]. As expected, we confirmed that WRKY41 and WRKY53 also directly bind to the *LFY* promoter (Figure 6C,E and Appendix A). These results suggest that *SOC1* and *LFY* are target genes of WRKY41 and WRKY53 and that high *AP1* expression may be caused by *LFY*. In genetic phenotype, studies have shown that the loss-functions of *SOC1* or *LFY*, such as *soc1-2* lines and *lfy-2* lines, delays flowering, and that *SOC1* or *LFY* overexpression leads to early flowering in *Arabidopsis* [21,66]. In our present study, the loss function of *WRKY41* or *WRKY53* both weakened the GV-induced early flowering (Figure 2F and Figure 3F), meanwhile, the early flowering phenotypes of *35S:WRKY41* lines were reversed in the *soc1* or *lfy* background (Figure 7C–F), implying that WRKY41 and WRKY53 accelerated flowering in a manner dependent on *SOC1*/*LFY*. Together, these observations provide supporting evidence that WRKY41 and WRKY53 co-mediate GV-accelerated flowering by directly activating the transcription of *SOC1* and *LFY* in *Arabidopsis*.

## 4. Materials and Methods 

### 4.1. Plant Materials and Growth Conditions

All *Arabidopsis* plants are in the Col background. Plants were grown under cool white fluorescent lights (80–100 μmol m^−2^ s^−1^) at 22 °C and 60% relative humidity in incubators (LEDIAN, Ningbo, China) [44]. The long-day (LD) conditions consisted of a 16 h light/8 h dark and short-day (SD) conditions consisted of an 8 h light/16 h dark photoperiod. Col-0 *Arabidopsis* was used as the wild type (WT). The T-DNA insertion knockout mutants *wrky41* (*Salk_068648*) and *wrky53* (*Salk_034157*) [67] were provided by the AraShare *Arabidopsis* Stock Centre (Fuzhou, China). Overexpression lines *35S:WRKY41*-GFP and *35S:WRKY53*-GFP were generated by cloning the full-length *WRKY41* and *WRKY53* coding sequence (CDS) into the pCHF3 vector, which contains a 35S promoter and a GFP-tag used for screening positive overexpressing plants by western blot and performing ChIP-qPCR assays with anti-GFP antibodies [68,69]. The following mutants were obtained by using CRISPR/cas9-mediated knockout technology [70,71], the *w41/w53-1* and *w41*/*w53-2* double mutants were generated by knocking out *WRKY53* in *wrky41* lines and *WRKY41* in *wrky53* lines, respectively. The 35*S:WRKY41*-*soc1* and 35*S:WRKY41*-*lfy* lines were generated by knocking out *SOC1* and *LFY*, respectively, in *35S:WRKY41* lines. The *soc1* and *lfy* lines were generated by knocking out *SOC1* and *LFY* in the WT. 

### 4.2. Generation of Transgenic Plants 

To construct overexpression lines, the corresponding gene CDSs for *WRKY41* and *WRKY53* were amplified and introduced into the pCHF3 vector [72]. To generate the knockout plants, CRISPR/Cas9-mediated genome editing was used. The target sites for *WRKY41* (Oligo1:5′-gattgtctcaacaaatacttccac-3′, Oligo2:5′-aaacgtggaagtatttgttgagac-3′), *WRKY53* (Oligo1:5′-gattggccattacccaaaagccaa-3′, Oligo2:5′-aaacttggcttttgggtaatggcc-3′), *SOC1* (Oligo1:5′-gattgagtgactttctccaaaaga-3′, Oligo2:5′-aaactcttttggagaaagtcactc-3′), and *LFY* (Oligo1:5′-gattgagacgattgcaagaagagg-3′, Oligo2:5′-aaaccctcttcttgcaatcgtctc-3′), respectively, were designed with CRISPR-P2.0 (http://crispr.hzau.edu.cn/CRISPR2/, accessed on 4 May 2023) and inserted into the pCAMBIA1300 vector. Constructs were transfected into the WT, *35S:WRKY41*, *wrky41*, and *wrky53* lines via floral dip with *Agrobacterium tumefaciens* strain (GV3101) [73]. Homozygous plants were identified by sequencing. All primers used were summarized in Appendix A.

### 4.3. Gene Expression Analysis

Two-week-old *Arabidopsis thaliana* leaves were sprayed with 50 mg L^−1^ guvermectin (GV) once, and the whole plant was ground with liquid nitrogen at 0, 1, 3, 5, and 7 d after GV treatment, then total RNA was extracted using TRIzol reagent [74]. Meanwhile, the plants sprayed with 0 mg L^−1^ GV were used as a control at each point. For qRT-PCR, 1 μg total RNA per sample was treated with the PrimeScript™ RT reagent Kit with gDNA Eraser (RR047A, Takara, San Jose, CA, USA). qRT-PCR was performed with SYBR Master Mix (Q711-03, Vazyme, Nanjing, China) on the Bio-Rad iQ5 optical system software (Bio-Rad, Hercules, CA, USA). UBQ5 was used as the internal control gene for expression level normalization. The transcript level of each gene was calculated using the double ΔCt method [34]. Data analysis was conducted in GraphPad Prism 8. The primers were summarized in Appendix A.

### 4.4. Transcriptome Analysis

RNA sequencing (RNA-seq) was performed according to Xie et al. [75], with minor modifications. Two-week-old *Arabidopsis thaliana* grown under LD conditions were treated by spraying with 50 mg L^−1^ GV once and the plants sprayed with 0 mg L^−1^ GV were used as a control. The whole plant, four days after GV treatment, was collected for RNA extraction and RNA-seq. Sequencing was performed on an Illumina NovaSeq 6000 (Illumine, Austin, TX, USA). Three independent biological replicates were sequenced and analyzed.

### 4.5. Electrophoretic Mobility Shift Assay (EMSA)

To investigate the interaction between the transcription factors *WRKY41* and *WRKY53* and the promoters of *SOC1* and *LFY*, The EMSA was performed as described previously [76,77] The CDSs of *WRKY41* and *WRKY53* were inserted into a vector containing the GST tag. GST-WRKY41 and GST-WRKY53 proteins were then expressed and purified using GST Protein Purification System (GE Healthcare, Chicago, IL, USA). The NDA Probes 200 bp (10 ng) generated based on the promoter sequences of *SOC1* and *LFY* and various concentrations of GST-WRKY41 and GST-WRKY53 were incubated in 20 μL binding buffer (20 mmol/L Tris base, 5% (*v*/*v*) glycerol, 2 mmol/L dithiothreitol, 5 mmol/L MgCl_2_, and 0.5 μg BSA) at 25 °C for 25 min. The reaction mixtures were analyzed with 4% non-denaturing polyacrylamide gel after electrophoretic at 4 °C. The DNA-protein complexes were observed after incubation with the addition of SYBR GOLD chemiluminescent dyes for 1 h and photographed under ultraviolet transillumination. GST was used as the negative control. The primers were summarized in Appendix A.

### 4.6. ChIP- qPCR Assay

Chromatin immunoprecipitation (ChIP) was performed as previously described [78,79], with minor modifications. Approximately 4 g of four-week-old plant tissue per sample was used for ChIP-qPCR analysis. The plant lines analyzed were *35S:WRKY41*-GFP, *35S:WRKY53*-GFP, and WT. The protein-DNA complexes of *35S:WRKY41*-GFP, *35S:WRKY53*-GFP, and WT lines were incubated with GFP-Trap Agarose Beads (ChromoTek). The enrichment of DNA fragments was determined by qPCR. Three independent biological replicates were performed. The primers were listed in Appendix A.

### 4.7. Dual-Luciferase Assay

DNA segments (2 kb in length) of the *SOC1* and *LFY* promoters were cloned and inserted into pGreenII 0800-Luc to activate the Luc reporter gene. This plasmid carries a Renilla luciferase gene (*REN*) as the internal control. The CDS of *WRKY41* and *WRKY53* were also inserted into pGreenII-62-SK to serve as effectors. Each vector was transformed into *A. tumefaciens* GV3101 (*pSoup*), and cells were transfected into the *N. benthamiana* after mixing in a ratio of 1:9. The activity of Luc and Ren were detected with a multifunctional microplate reader at 48 h after transfection, respectively. The ratio of Luc:Ren was calculated as previously described [67,72,80] with minor modifications. The primers were listed in Appendix A.

### 4.8. The Phylogenetic Construction

The phylogenetic construction is according to WU et al. [50]. The *Arabidopsis* WRKY genes were obtained in NCBI GenBank (https://www.ncbi.nlm.nih.gov/gene/, accessed on 4 May 2023), and then the phylogenetic tree was constructed using Software MEGA6.0.6.

## 5. Conclusions

In summary, our results confirmed the biological activity of guvermectin (GV) in accelerating flowering. The expression of transcription factors *WRKY41* and *WRKY53* were significantly induced by GV, suggesting they may play a role. The phylogenetic tree results indicated that *WRKY53* is the closest homolog to *WRKY41*. Overexpression of *WRKY41* or *WRKY53* lines showed an early flowering phenotype and the double knockout mutants *w41/w53* lines showed a late-flowering phenotype in *Arabidopsis*, indicating *WRKY41* and *WRKY53* play an important role in regulating flowering. Meanwhile, after GV treatment, *wrky41* or *wrky53* lines showed a weakened early flowering than WT and *w41/w53* lines showed a late-flowering phenotype, which provided evidence that *WRKY41* and *WRKY53* jointly mediate GV-accelerated flowering. Gene expression analysis showed GV induced the expression of *SOC1* and *LFY* through WRKY41 and WRKY53. Furthermore, we confirmed that *SOC1* and *LFY* are both direct targets of WRKY41 and WRKY53. Together, these results revealed that WRKY41 and WRKY53 co-mediate GV-accelerated flowering by directly activating the transcription of *SOC1* and *LFY* in Arabidopsis (Figure 8).

## Figures and Tables

**Figure 1 ijms-24-08424-f001:**
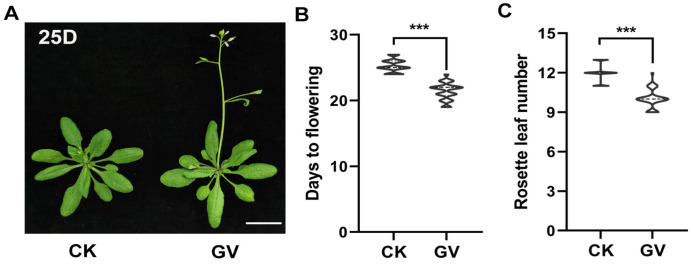
Early flowering phenotype induced by GV in *Arabidopsis*. (**A**) Representative images showing the flowering phenotypes of CK (treated with 0 mg L^−1^ GV) plants and those treated with 50 mg L^−1^ GV in *Arabidopsis* grown under long-day (LD) conditions. (**B**,**C**) Flowering phenotypes associated with GV treatment and CK, as assessed by DTF (**B**) and RLN (**C**), were grown under LD conditions. Two-week-old plants were sprayed with 50 mg L^−1^ GV and CK, and the DTF and RLN were assessed, respectively. 25D: DTF of wild type (WT) plants grown under LD conditions. CK, control (treated with 0 mg L^−1^ GV). GV, guvermectin treatment. Three biological replicates were counted with similar results. Values are expressed as means ± SD (*n* = 30). A significant difference analysis was the Student’s *t*-test (***, *p* < 0.001). Bar = 1 cm.

**Figure 2 ijms-24-08424-f002:**
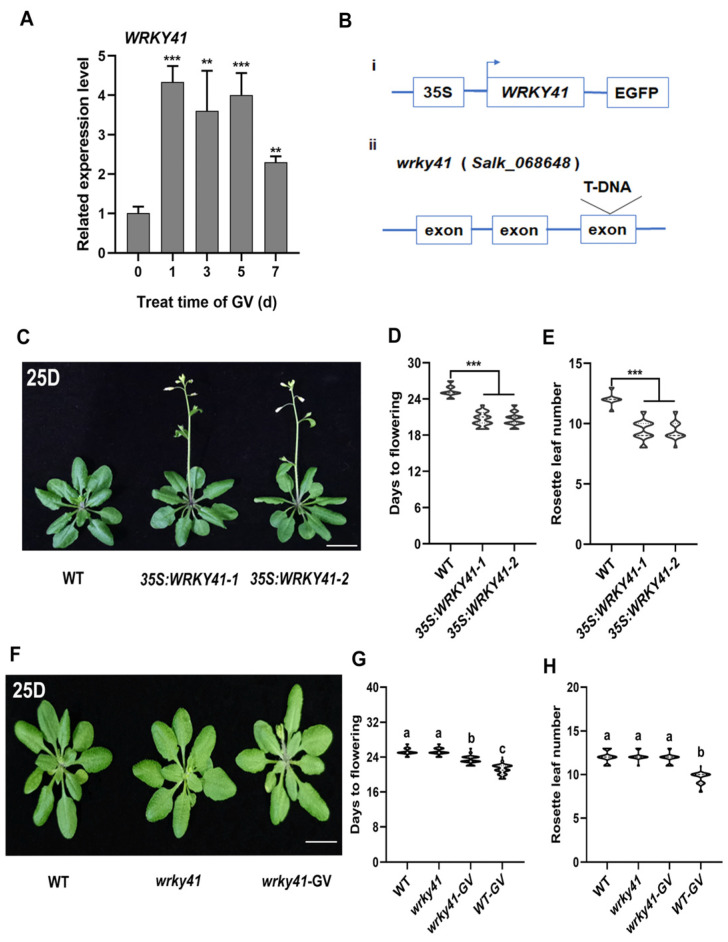
*WRKY41* plays a role in GV-induced early flowering. (**A**) The related transcript levels of *WRKY41* at 0, 1, 3, 5, and 7 d after GV treatment. A significant difference analysis was the Student’s *t*-test (**, *p* < 0.01; ***, *p* < 0.001). (**B**) i: *WRKY41* overexpression, the *WRKY41* CDS was inserted into a vector with 35S promoter; ii: The T-DNA insertion site in the *wrky41* knockout line; the insertion site was in an exon. (**C**) Representative images of flowering phenotypes in WT and two *35S:WRKY41* lines. (**D**,**E**) The flowering phenotypes of 35S:*WRKY41* and WT plants were assessed using DTF (**D**) and RLN (**E**). A significant difference analysis was the Student’s *t*-test (***, *p* < 0.001). (**F**) Representative images of the flowering phenotypes for WT, and *wrky41* plants with or without GV treatment. (**G**,**H**) Flowering phenotypes of WT and *wrky41* plants with or without GV treatment were assessed by DTF (**G**) and RLN (**H**). *wrky41*, *wrky41* plants treated with control (treated with 0 mg L^−1^ GV). *wrky41*-GV, *wrky41* plants treated with 50 mg L^−1^ GV. 25D, DTF for WT under LD conditions. Different letters above the bars indicate statistically significant differences (adjusted *p* < 0.05, one-way ANOVA). All experiments were repeated three times with similar results. Values are expressed as means ± SD (*n* = 30). Bar = 1 cm.

**Figure 3 ijms-24-08424-f003:**
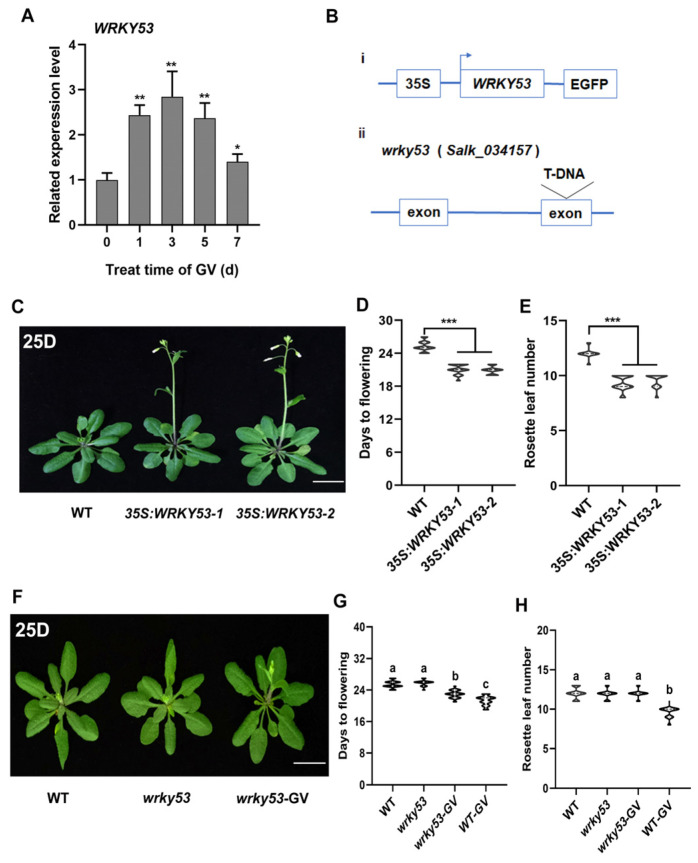
*WRKY53* plays a role in GV-induced early flowering. (**A**) The related transcript levels of *WRKY53* at 0, 1, 3, 5, and 7 d after GV treatment. A significant difference analysis was the Student’s *t*-test (*, *p* < 0.05; **, *p* < 0.01). (**B**) i: *WRKY53* overexpression; the *WRKY53* CDS was inserted into a vector with 35S promoter; ii: The T-DNA insertion site in the *wrky53* knockout line; the insertion site was in an exon. (**C**) Representative images of the flowering phenotypes of WT and *35S:WRKY53* lines. (**D**,**E**) Flowering phenotypes of WT and two *35S:WRKY53* lines were assessed by DTF (**D**) and RLN (**E**). A significant difference analysis was the Student’s *t*-test (***, *p* < 0.001). (**F**) Representative images of the flowering phenotypes for WT, and *wrky53* lines treatment without and with GV. (**G**,**H**) The flowering phenotype of WT and *wrky53* plant treatment with and without GV treatment were assessed by DTF (**G**) and RLN (**H**). *wrky53*, *wrky53* plants treated with control (treated with 0 mg L^−1^ GV). *wrky53*-GV, *wrky53* plants treated with 50 mg L^−1^ GV. 25D, DTF for WT and *wrky53* lines. Different letters above the bars indicate statistically significant differences (adjusted *p* < 0.05, one-way ANOVA). All experiments were repeated three times with similar results. Values are expressed as means ± SD (*n* = 30). Bar = 1 cm.

**Figure 4 ijms-24-08424-f004:**
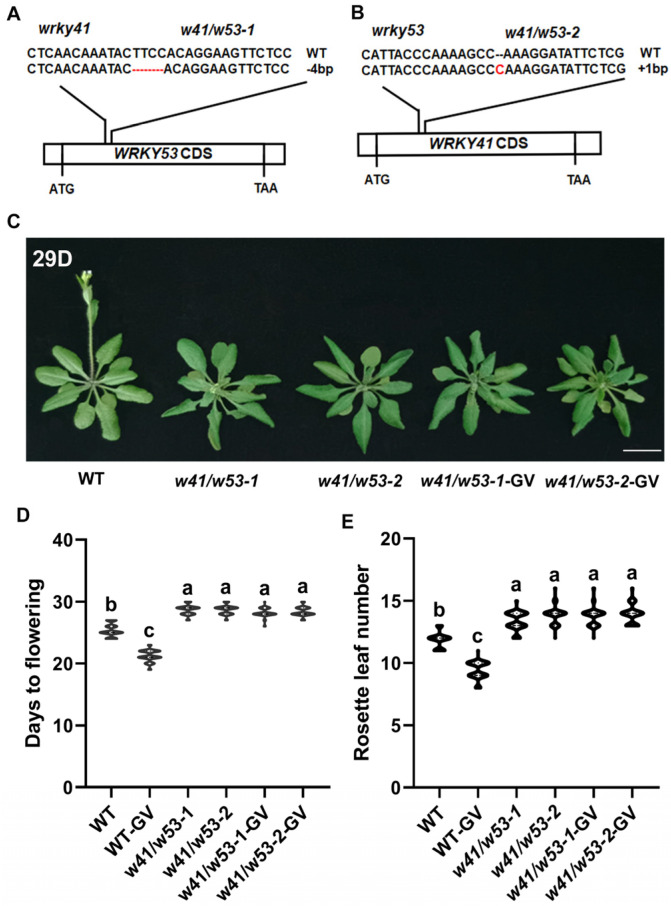
Flowering is delayed in *w41*/*w53* mutants. (**A**,**B**) Construction of CRISPR/Cas9-based *w41/w53* knockout transgenic lines. Single guide RNAs (sgRNAs) were generated to target *WRKY53* in *wrky41* lines (**A**) and *WRKY41* in *wrky53* lines (**B**), respectively, to generate two double mutant lines, *w41/w53-1* and *w41/w53-2*. The sequence ‘TTCC’ was deleted from *WRKY41* in *w41/w53-1* and a red ‘C’ was inserted into *WRKY53* in *w41/w53-2*. (**C**) Representative images of the flowering phenotypes of WT and *w41/w53* plants. (**D**,**E**) Flowering phenotypes of WT and *w41/w53* plants with and without GV treatment were assessed by DTF (**D**) and RLN (**E**). 29D, DTF of *w41/w53* lines under LD conditions. Different letters above the bars indicate statistically significant differences (adjusted *p* < 0.05, one-way ANOVA). Three biological replicates were counted with similar results. Values are expressed as means ± SD (*n* = 30). Bar = 1 cm.

**Figure 5 ijms-24-08424-f005:**
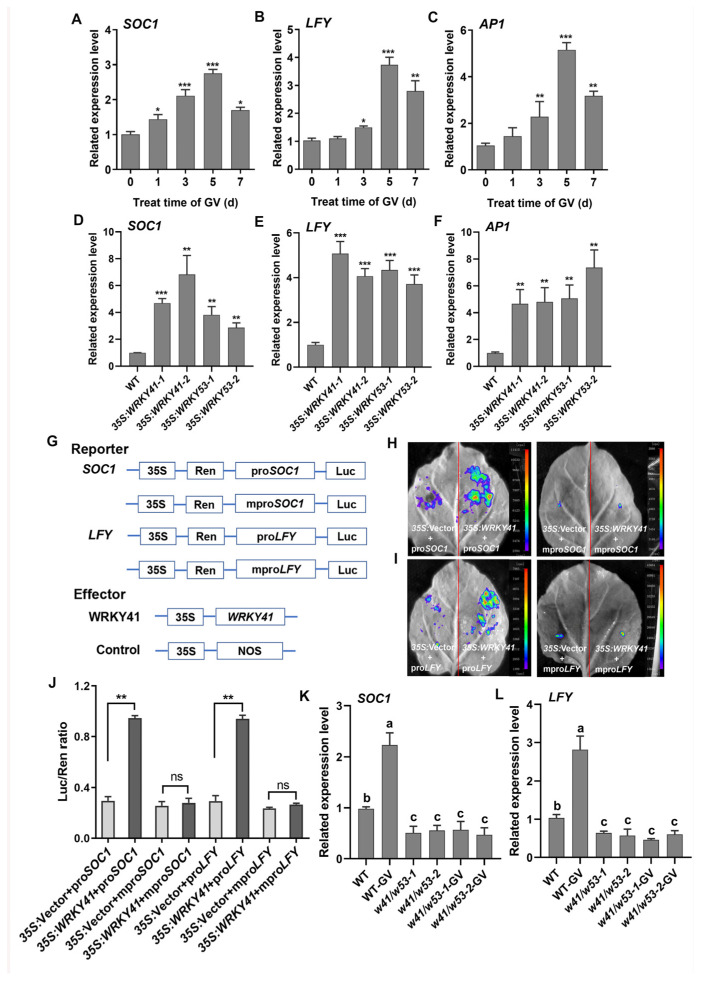
WRKY41 and WRKY53 activate critical flowering regulatory genes. (**A**–**C**) The related expression levels of the key flowering regulatory genes *SOC1* (**A**), *LFY* (**B**), and *AP1* (**C**) responsed to GV treatment. The samples were collected at 0, 1, 3, 5, and 7 d after the treatment of 50 mg L^−1^ GV and control (treated with 0 mg L^−1^ GV). A significant difference analysis was the Student’s *t*-test (*, *p* < 0.05; **, *p* < 0.01; ***, *p* < 0.001). (**D**–**F**). Related expression levels of *SOC1* (**D**), *LFY* (**E**), and *AP1* (**F**) are shown in the *35S:WRKY41* and *35S:WRKY53* lines. A significant difference analysis was the Student’s *t*-test (**, *p* < 0.01; ***, *p* < 0.001). (**G**) The constructs were used for the transient transcriptional activity assay. The native and mutant promoters of *SOC1* and *LFY* were used as reporters, respectively. WRKY41 was used as an effector. 35S, CaMV35S promoter. Luc, firefly luciferase. Ren, Renilla luciferase. (**H**) Transcription activation detection between WRKY41 and the pro*SOC1* (**H** left) and mpro*SOC1* (**H** right). (**I**) Transcription activation detection between WRKY41 and pro*LFY* (**I** left) and mpro*LFY* (**I** right). WRKY41 activated the expression of luciferase driven by the *SOC1* and *LFY* promoters. (**J**) Luc:Ren ratio after WRKY41 activating *SOC1* and *LFY* transcription. pro*SOC1*/pro*LFY*, the native promoter of *SOC1*/*LFY*. mpro*SOC1*/mpro*LFY*, the mutant promoter of *SOC1*/*LFY*. A significant difference analysis was the Student’s *t*-test (**, *p* < 0.01, ns, not significant). (**K**,**L**) Relative expression levels of *SOC1* (**K**) and *LFY* (**L**) in WT and *w41/w53* lines treated with 0 mg L^−1^ and 50 mg L^−1^ GV, respectively. WT, *w41/w53-1/-2* treated with 0 mg L^−1^ GV. WT-GV, *w41/w53-1/-2-*GV treated with 50 mg L^−1^ GV. Different letters above the bars indicate statistically significant differences (adjusted *p* < 0.05, one-way ANOVA). Each experiment was repeated three times with similar results. Values are expressed as means ± SD (*n* = 3).

**Figure 6 ijms-24-08424-f006:**
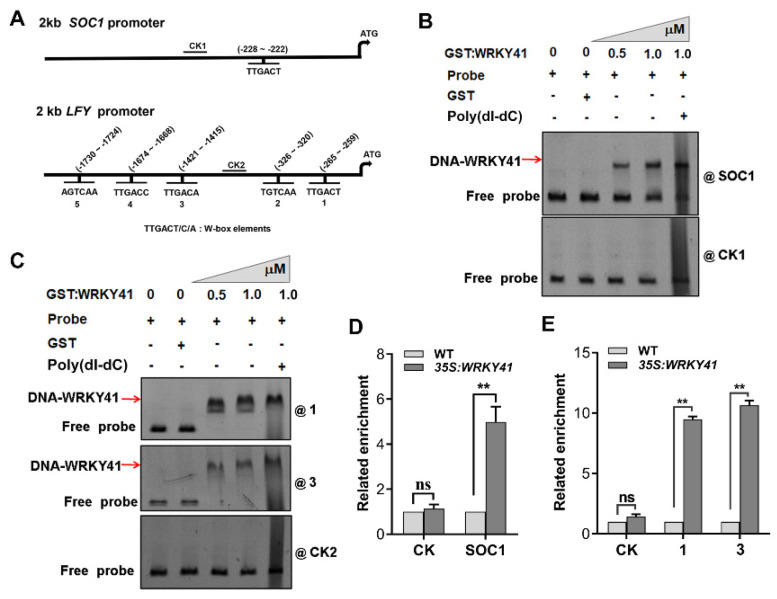
WRKY41 directly binds the *SOC1* and *LFY* promoters. (**A**) The 2-kb promoter regions and fragments of *SOC1* and *LFY* were used in electrophoretic mobility shift assays. (−228~−222), The position of the W-box in the *SOC1* promoter. 1, 2, 3, 4, and 5 represent the W-boxes located at (−265~−259), (−326~−320), (1421~−1415), (−1674~−1668), and (−1730~−1724) bp in the *LFY* promoter. (**B**,**C**) EMSA showed that GST-WRKY41 directly binds to the W-box in the promoter at (−228~−222) bp in the *SOC1* promoter (**B**) and (−265~−259) bp (1) and (1421~−1415) bp (3) in the *LFY* promoter (**C**). 100-fold non-specific poly(dI-dC) was used to exclude non-specific binding between protein and probes. CK1 and CK2, negative control. The plus (+) and minus (−) indicate the presence and absence of the indicated components. Arrows indicate band shifts. The triangle symbol indicates an increased concentration of GST-WRKY41. (**D**,**E**) Enrichment of the W-box in the *SOC1* promoter (**D**) and W-box-1 and W-box-3 in the *LFY* promoter (**E**) based on ChIP-qPCR. Samples were collected from three-week-old *35S:WRKY41* plants. All experiments were repeated three times with similar results. Values are expressed as means ± SD (*n* = 3). A significant difference analysis was the Student’s *t*-test (ns, not significant; **, *p* < 0.01).

**Figure 7 ijms-24-08424-f007:**
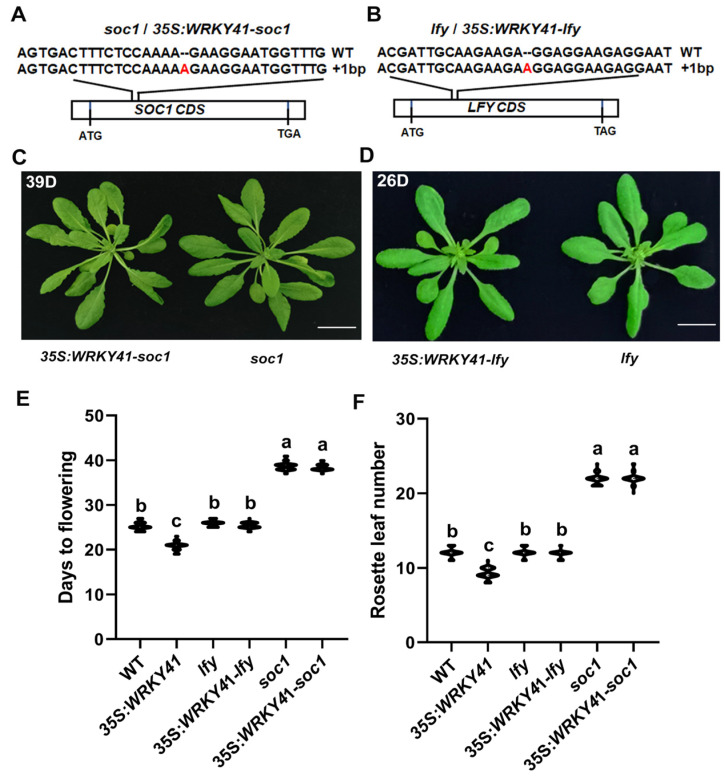
WRKY41 directly activates *SOC1* and *LFY* to promote flowering in *Arabidopsis*. (**A**,**B**) Construction of the *35S:WRKY41-soc1* and *soc1* (**A**) and the *35S:WRKY41-lfy* and *lfy* (**B**) knockout lines via CRISPR/Cas9. Single guide RNAs (sgRNAs) targeting *SOC1* or *LFY* were used in the *35S:WRKY41* and WT lines to generate four mutant lines (*35S:WRKY41-soc1*, *soc1*, *35S:WRKY41-lfy,* and *lfy*). A red ‘A’ was inserted into *SOC1* in the *35S:WRKY41* and WT lines and a red ‘A’ was inserted into *LFY* in the *35S:WRKY41* and WT lines. (**C**,**D**) Representative images showing flowering phenotypes of *35S:WRKY41-soc1* and *soc1* (**C**), and *35S:WRKY41-lfy* and *lfy* (**D**). (**E**,**F**) Flowering phenotypes of WT, *35S:WRKY41*, *35S:WRKY41-soc1*, *soc1*, *35S:WRKY41-lfy* and *lfy* lines as assessed by DTF (**E**) and RLN (**F**). Values are expressed as means ± SD (*n* = 15). Similar results were obtained from three biological repeats. Different letters above the bars indicate statistically significant differences (adjusted *p* < 0.05, one-way ANOVA). Bar = 1 cm.

**Figure 8 ijms-24-08424-f008:**
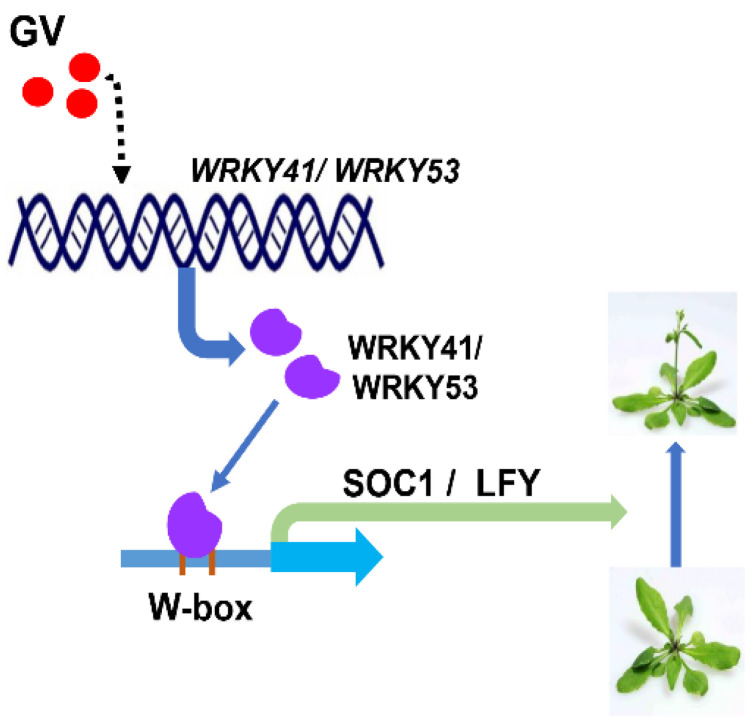
A proposed model illustrating early flowering induced by GV. In the presence of GV, transcription factors *WRKY41* and *WRKY53* are induced to express. Then, they directly activate the transcripts of two key flowering regulation genes *SOC1* and *LFY* by binding to their promoters, respectively. At the same time, the expression levels of *SOC1* and *LFY* increase, resulting in early flowering in *Arabidopsis*. *WRKY41*/*WRKY53*, gene *WRKY41* and *WRKY53*. WRKY41/WRKY53, protein WRKY41 and WRKY53. SOC1/LFY, protein SOC1 and LFY, W-box, the W-box elements in the promoters. The dotted arrow line represents indirect action and the solid arrow line represents direct action.

## Data Availability

Data supporting the findings of this work are available within the paper and its Appendix A. RNA-Sequence data have been deposited in the Sequence Read Archive (SRA) (https://www.ncbi.nlm.nih.gov/sra/, accessed on 4 May 2023) with the accession number PRJNA851570.

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
