# Peer review of "The Transcription Factors WRKY41 and WRKY53 Mediate Early Flowering Induced by the Novel Plant Growth Regulator Guvermectin in Arabidopsis thaliana"

_ijms, 2023, doi:10.3390/ijms24098424_

Round 1

Reviewer 1 Report

The manuscript by Yang et al. reports the identification of guvermectin (a novel plant growth regulator) as a floral activator through WRKY41 and WRKY53 TF, by activating the expression of floral integrator genes, SOC1 and LFY.

The authors confirm the finding using genetic analysis by generating mutants (via CRISPR) and overexpression plants. Furthermore, the author also demonstrated the role of WRKY41 and WRKY53 as a direct transcription activator of SOC1 and LFY using EMSA, ChIP, and luciferase effector-reporter assay. Overall, the manuscript is well-written, and the experiments have been executed well. However, the weak flowering time phenotype of guvermectin treatment make the finding seems not that convincing.

Major comment:

1. While the author shows that guvermectin (GV) application significantly accelerated flowering, the effect is relatively weak (probably just 1-2 leaf difference). The statistically significant difference observed in most of the flowering time graphs (ex. 1C, 2E, 2H, 3E, 3H, 4E) is likely caused by the sample size/effect size. 

In my opinion, the authors should show more convincing data that GV and WRKY41/53 indeed have a function to promote flowering, by: 1. Increasing the concentration of GV treatment, 2. Measure the flowering time at non-inductive short-day (SD) conditions. 3. Use total leaves (rosette + cauline) for flowering time indicator (see Pouteau and Albertini, 2009). Considering that GV acts upstream of SOC1 and LFY, the effect of GV should be much stronger at SD instead of LD. Furthermore, LFY is primarily acting in the 2nd stage of floral initiation; hence the lfy mutants have much more cauline leaves than WT, while the number of rosette leaves is relatively similar.

Furthermore, all graph (especially for the flowering time) is better presented with data points. The author also can use a box/violin-plot to show the data distribution. Presenting data points/graph distribution will help the reader to discern the biological effect of the treatments.

2. I am concerned about the generation of soc1 and lfy mutants. The author independently generated soc1/lfy mutants in wild type and WRKY41ox lines by CRISPR. I assume the author chooses crispr lines with similar adenine insertion in both backgrounds (Fig. 7A and B). However, this approach cannot rule out the effect of different off-targets, especially since the author seems didn’t select any T-DNA free lines for subsequent analysis. 

Figure 7 also lacks WT and WRKY41ox control. 

3. The EMSA and ChIP data show convincing results that WRKY41/53 recognize and bind to the W-box element in SOC1 and LFY promoter. However, this conclusion could be supported more by additional experiments using luciferase assay with mutated W-box promoters.

Minor comment:

1. line 111-128 is still in italics.

2. The author should have a summary of flowering time data in the table (see Table 1 from Xu et al. 2016 or Hou et al. 2014). The table can be presented as supplemental data.

3. No scale bar in all plant photos.

4. The author can improve the introduction by adding more information related to guvermectin.

Reference

Hou, X., Zhou, J., Liu, C. et al. Nuclear factor Y-mediated H3K27me3 demethylation of the SOC1 locus orchestrates flowering responses of ArabidopsisNat Commun 5, 4601 (2014). https://doi.org/10.1038/ncomms5601

Sylvie Pouteau, Catherine Albertini, The significance of bolting and floral transitions as indicators of reproductive phase change in ArabidopsisJournal of Experimental Botany, Volume 60, Issue 12, August 2009, Pages 3367–3377, https://doi.org/10.1093/jxb/erp173

Xu M, Hu T, Zhao J, Park M-Y, Earley KW, Wu G, et al. (2016) Developmental Functions of miR156-Regulated SQUAMOSA PROMOTER BINDING PROTEIN-LIKE (SPL) Genes in Arabidopsis thaliana. PLoS Genet 12(8): e1006263. https://doi.org/10.1371/journal.pgen.1006263

Reviewer 2 Report

The authors investigated effects of guvermectin on flowering time regulation in Arabidopsis. Overall, the manuscript was well structured and their claims were appropriately supported by data, but some points need to be addressed before publication. 

1. The author need to compare the effect of guvermectin (GV) with cytokinin on flowering time regulation. For example, D’Aloia et al (2011) reported that cytokinin (BAP) accelerates floral initiation in Arabidopsis through transcriptional activation of TSF and SOC1. Based on this information, the authors need to compare the effect of GV and cytokinin on flowering time regulation.

D'Aloia M, Bonhomme D, Bouché F, Tamseddak K, Ormenese S, Torti S, Coupland G, Périlleux C. Cytokinin promotes flowering of Arabidopsis via transcriptional activation of the FT paralogue TSF. Plant J. 2011 Mar;65(6):972-9. doi: 10.1111/j.1365-313X.2011.04482.x. Epub 2011 Feb 16. PMID: 21205031.

2. In addition, it is important to analyze effect of GV treatment on flowering time regulation under non-inductive conditions. 

3. Line 62-66: The authors introduced function of WRKY from different species. Some genes have species information but not others. This need to be revised.

4. Line 111 to 128. Please revised the entire paragraph. Now all the characters in the paragraph are italic.

Figure 2,3,7: Please provide expression level of WRKY41(or 53) in the transgenic plants.

5. They used 35S:WRKY41/53-GFP line as a overexpressor. Please rationalize why the author did not generated 35S:WRKY plants.

6. Figure 4(C,D,E) and 5. please add data of WT treated with GV for comparison.

7. Please add scale bar for the figures.

Round 2

Reviewer 1 Report

This revision has significantly improved the manuscript quality. I have no further comments.

Reviewer 2 Report

The authors appropriately responded to the comment with additional data and corresponding descriptions in the revised version.